# Intrahepatic Cholangiocarcinoma Identified in a Zoo-Housed Sandhill Crane (*Grus canadensis*): An Anatomopathological and Metagenomic Study

**DOI:** 10.3390/ani13223469

**Published:** 2023-11-10

**Authors:** Hye-Ryoung Kim, Hyeon-Su Kim, Yong-Kuk Kwon

**Affiliations:** Avian Disease Division, Animal and Plant Quarantine Agency, 177 Hyeoksin 8-ro, Gimcheon-si 39660, Gyeongsangbuk-do, Republic of Korea; stevenkim0401@korea.kr (H.-S.K.); kwonyk66@korea.kr (Y.-K.K.)

**Keywords:** intrahepatic cholangiocarcinoma, sandhill crane, birds, neoplasm, metagenomics

## Abstract

**Simple Summary:**

A male sandhill crane (*Grus canadensis*) was imported from the Netherlands in 2013 and bred for 8 years in a zoo for a crane-species restoration project in Gangwon Province in South Korea. The adult bird showed no clinical symptoms before its death. This study describes the anatomopathological finding of intrahepatic cholangiocarcinoma diagnosed in the captive sandhill crane, and metagenomic sequencing was performed to elucidate the etiology. This is the first report where the gross and histological characteristics of intrahepatic cholangiocarcinoma in a sandhill crane are described. No viral cause was identified in the metagenomics analysis and PCR of the cholangiocarcinoma sample, but a contributing role of *Cutibacterium* sp. and *E. coli* identified from the metagenomics could not be excluded.

**Abstract:**

Tumors in birds can be caused by a variety of factors such as species, age, sex, virus, chemicals, and environment. In particular, tumors are a major cause of death in long-lived birds such as parrots and zoo birds. A male sandhill crane that was bred for 8 years in a zoo was diagnosed with intrahepatic cholangiocarcinoma (ICC). At necropsy, the liver revealed a multinodular mass of variable colors, and severe cirrhosis and hemorrhages were present. Histologically, ICC was characterized by the presence of both types of ICC: small-duct type and large-duct type. Large-duct-type ICC was distinguished by the presence of multifocal biliary neoplasia, characterized by the diffuse papillary proliferation of columnar cells resembling large cholangiocytes. Small-duct-type ICC was characterized by the presence of non-mucin-producing cuboidal cells such as bile duct cells. In this case, no viral cause was identified from the metagenomic analysis and PCR of ICC; however, a contributing role of *Cutibacterium* sp. and *E. coli* identified from the metagenomics could not be excluded. This study is the first to describe the anatomopathological characteristics of ICC in the studied sandhill crane and attempts to determine its potential infectious etiology using metagenomics.

## 1. Introduction

The sandhill crane (*Grus canadensis*) is a species of bird that belongs to the *Gruidae* family, with population distributed widely across North American, including Mexico, and extending to Russia’s Far East. They are migratory birds and even occur in China during winter. Their populations were destroyed by habitat loss and hunting in the 1930s, but recovered in 2018, and they are not considered threatened as a species, except for some subspecies. These birds can live for 20 years or more in the wild [1,2,3].

There are a variety of diseases, including Newcastle disease, highly pathogenic avian influenza, avian pox, avian cholera, and avian botulism, along with habitat loss, that are life-threatening factors for wild birds [4]. Sandhill cranes can be naturally infected with infectious bursal disease, have been infected with avian pox and Eimerian Coccidia, and have been killed by mycotoxin-induced disease and lead poisoning. Migratory flocks are potentially susceptible to outbreaks of avian cholera and other diseases under low-flow conditions. [3,5,6,7,8]. In particular, tumors may also be included as a major cause of death in long-lived birds of the genus Grus [9,10]. Tumors in birds can be caused by a variety of factors, such as species, age, sex, virus, chemicals, and environmental factors. There are many studies on typical tumors induced by avian oncogenic viruses, such as Marek’s disease virus (MDV), avian leukosis virus (ALV), and reticuloendotheliosis virus (REV), in poultry, but other tumors of unknown etiology are reported intermittently [11].

This study presents the anatomopathological findings of intrahepatic cholangiocarcinoma (ICC) diagnosed in a captive sandhill crane. In addition, metagenomic sequencing was performed to elucidate its etiology.

## 2. Materials and Methods

### 2.1. History

The carcass of a male, adult, captive sandhill crane from a zoo located in Gangwon Province in South Korea was submitted to postmortem examination and diagnostic work-up at the Avian Disease Division of the Animal and Plant Quarantine Agency (APQA) in November 2021. The bird, which was imported from the Netherlands in 2013 and bred for 8 years in a zoo, belonged to a crane-species restoration project in Gangwon Province. The bird showed no clinical symptoms before its death.

### 2.2. Sample Collection and High-Throughput Sequencing

Necropsy, virus isolation using specific pathogen-free (SPF) embryonated chicken eggs, and histological examination were performed according to an APQA diagnostic protocol. Allantoic fluid obtained from SPF embryonated eggs inoculated with aliquots of homogenized trachea, cecal tonsil, and kidney were tested against avian influenza virus and Newcastle disease virus via hemagglutination tests. In addition, the trachea, cecal tonsil, kidney, and liver were homogenized in phosphate-buffered saline (PBS), and viral DNA was extracted using a QIAamp DNeasy Blood & Tissue kit (Qiagen, Hilden, Germany). MDV and REV were screened using a BlackCheck CIAV/MDV/REV Multi Detection Kit (Ventech Science, Daegu, South Korea). PCR was performed to detect ALV, as described previously [12]. No culture for bacteriologic studies was performed.

Sections of the organs (trachea, cecal tonsil, and kidney), including the liver with distinct lesions, were collected and fixed in 5% buffered formalin. The samples were embedded in paraffin blocks, and paraffin wax sections were cut (5 µm), dewaxed, stained with hematoxylin and eosin, and examined using light microscopy.

To identify a potential etiologic pathogen, a liver sample of the bird was collected after necropsy and promptly processed via blending into a 10% homogenate in sterile phosphate-buffered saline containing 0.4 mg/mL gentamicin. A liver sample from a healthy chicken was used as negative control for the sample pretreatment and sequencing. The collected supernatant was subsequently filtered and was added to 8% polyethylene glycolv6000 and 0.5M NaCl as previously described [13,14]. The DNA/RNA of the resuspended pellet was extracted using a Maelstrom 4000 DNA/RNA auto-extraction machine (TAN bead, Taoyuan, Taiwan). First-strand cDNA synthesis was performed using a PrimeScript™ 1st strand cDNA Synthesis Kit (Takara, Shiga, Japan) according to the manufacturer’s instructions. Viral DNA and RNA were randomly amplified using an FR26-RV primer, as previously described [15], with LongAmp^®^ Taq 2X Master Mix (NEB, Ipswich, MA, USA). The amplified PCR fragments were purified using Agencourt AMPure XP beads (Beckman Coulter, Brea, CA, USA) at a 1.8x volume. The quantification of the purified dsDNA was performed via a dsDNA HS assay (Thermo Fisher Scientific, Waltham, MA, USA) using a Qubit fluorometer (Invitrogen, Waltham, MA, USA). An Illumina library was prepared using a Nextera XT DNA Library Preparation kit (Illumina, San Diego, CA, USA) and quantified using a Qubit fluorometer (Invitrogen, Waltham, MA, USA) and 4150 TapeStation system (Agilent Technologies, Santa Clara, CA, USA). Libraries were denatured and sequenced via Illumina MiniSeq using a MiniSeq Reagent kit (Illumina, San Diego, CA, USA).

### 2.3. Bioinformatic Analysis

The extracted dataset of paired-end reads was processed using Cutadapt (V2.8) and Trimmomatic (V0.39) to remove adapter, primer, and homopolymer sequences and shorter reads than 100 bp [16,17]. The remaining qualified reads were assembled into contigs using metaSPAdes (V3.15.4) [18]. To remove crane host reads, the contigs were mapped to the reference genome, *Grus japonensis* (GCA_002002985), *Grus gmonacha* (GCA_012487855), and *Grus gvipio* (GCA_012488435), using the Bowtie2 (V2.4.5) tool. The contigs were used as an index to align the qualified reads and a magnitude dataset was created to count the number of reads for the contigs. Finally, the contigs were classified using the Kraken2 (V2.1.2) program [19] with the NCBI nt database and graphically represented using a Krona Chart.

## 3. Results

### 3.1. Gross Findings and Histopathology

At necropsy, cachexia, dehydration, and a moderate volume of ascitic fluid were observed, but there was no distended abdomen. Right ventricular dilatation of the heart was evident, and heart failure may be considered the cause of death in this bird. The liver revealed a multinodular mass of variable colors ranging from white to yellow to dark red. The nodules were variable in size, solid, and not cystic. Severe cirrhosis and hemorrhages on the liver were also present. The left lobe was separated into three nodules and had a slightly firmer consistency compared to the right lobe. Multifocal hemorrhagic lesions were observed on the hepatic capsule. The size of liver that appeared to be a neoplasm was about 12 cm in diameter and this hepatic neoplasia mainly replaced the liver, and no remaining liver area was noted (Figure 1).

Histopathologic lesions in internal organs were found only in the liver. Two compartments can be seen, one area with neoplastic cells similar to the bile duct epithelium and the other area with a solid columnar-cell neoplasm containing only small luminal slits admixed with vacuolated hepatocytes and hemorrhages. Both parts of the neoplasm were separated by a thick fibrous pseudocapsule showing an incomplete capsule of variable thickness. In addition, perihepatitis, focal hepatic necrosis, and the infiltration of heterophils and lymphocytes were observed. Intralesional bacterial colonies were not observed histologically (Figure 2A). Large-duct-type ICC was characterized by mucin-producing papillary tumors caused by the diffuse papillary proliferation of bile duct cells without invasion or metastasis. Biliary neoplasia was composed of papillae covered by columnal cells forming dilated and distorted ducts resembling large cholangiocytes (Figure 2B). The other area was moderately differentiated small-duct-type ICC. Prominent and irregular ductular patterns of neoplastic cells infiltrated the adjacent hepatic parenchyma. Fibrin deposition, severe hemorrhage, and hemosiderosis are shown (Figure 2C). Small-duct-type ICC was characterized by non-mucin-producing cuboidal cells, such as bile ducts, with hyperchromatic nuclei and condensed cytoplasm, and the deposition of fibrin between tumor cells, replacing normal hepatocytes (Figure 2D). Columnar and mucin-producing cells of large-duct-type ICC were characterized by large nuclei and abundant eosinophilic cytoplasm (Figure 2E). Large-duct-type ICC showed a high mitotic index (6 to 9 per 10 high-powered field), but the mitotic index of small-duct-type ICC was 4 per 10 HPF or less. Both types of ICC, small and large-duct type, were diagnosed in the male adult sandhill crane.

### 3.2. Virus Identification

The allantoic fluid samples were negative in the hemagglutination test, and the presence of MDV, REV, and ALV was investigated via PCR, but no virus was detected.

### 3.3. Metagenomics Analysis

In order to search for pathogens that can affect ICC, a liver sample was sequenced, yielding a total of 13,851,588 reads. Of these, homology-based (BLAST) classification showed host reads (64.8%), bacteria (16.8%), eukaryota (1.8%), virus (<0.1%), others (0.2%) and unidentified reads (16.4%). In total, 17.7% (*n* = 388,437) and 13.1% (*n* = 288,422) were assigned to bacterial families comprising *Cutibacterium acnes* (family *Propionibacteriaceae*) and *Escherichia coli* (family *Enterobacteriacene*), respectively (Table 1). A total of 53.9% of bacterial reads were made up of a wide variety of other bacteria. The majority of viral sequences were classified as bacteriophages (98.5%) within the family Siphoviridae (*n* = 4154). Equine infectious anemia (*n* = 58) and fowl aviadenovirus E (*n* = 4) were found at much lower abundances. A total of 14,527,223 reads were obtained in the control sample, with a dominance of host reads. Most of the reads in the ICC samples were *Cutibacterium acne* and *E. coli*, which were not found in the control (Table 1).

## 4. Discussion

Neoplasia in *Gruidae* species was found to be relatively frequent compared to other avian orders, and most tumors identified in cranes were malignant [9]. Adenocarcinoma and hemangiosarcoma, adenoma, lymphoma, and cholangiocarcinoma in sandhill cranes have been reported, but the histopathological characteristics of the tumors have rarely been detailed previously [9,10]. This case report is the first to present the anatomopathological lesions of a typical malignant hepatic neoplasm, namely an ICC, in a sandhill crane.

Primary liver carcinomas are classified into the categories of hepatocellular carcinoma (HCC) and cholangiocarcinoma. HCC cases in zoo birds, such as the rosy-billed duck (*Netta peposaca*), red-headed duck (*Aythya americana*)*,* orange-headed ground thrush (*Zoothera citrina*), and Amazon parrot (*Amazona* species), as well as a wild-caught lesser flamingo (*Phoenicopterus minor*) and domestic ducks (*Anas platyrhynchos domestica*), have been reported in several studies, and a few cases of cholangiocarcinoma in domestic ducks (*Anas platyrhynchos domestica*) and a wild-caught Chilean flamingo (*Phoenicopterus ruber chilensis*) have been reported previously [20,21,22,23]. This case did not cover HCC, in which neoplastic cells differentiate towards hepatocytes, but ICC, where neoplastic cells arise from the peripheral bile duct, and bile ducts proliferate invasively, as evidenced by histopathological findings. The histopathologic description of this case followed the classification system of humans, due to the absence of a classification adapted to avian tumors. Human ICC is histopathologically classified into small-duct-type and large-duct-type ICC. Small-duct-type ICC is composed of non-mucin-producing cuboidal cells, whereas large-duct-type ICC is composed of mucin-producing columnar cells. Immunohistochemical markers are useful for differentiating small-duct-type and large-duct-type ICC. Mucin (MUC) core protein 5AC, MUC6, and S100 calcium-binding protein P are highly expressed in the large-duct type, but neural cell adhesion molecule and N-cadherin are highly expressed in the small-duct type. Large-duct-type ICC shows higher invasiveness and a poorer prognosis than the small-duct type [24,25]. Small-duct-type ICC occurs in intrahepatic bile ductules, and the large-duct type might be derived from biliary epithelium producing mucin or peribiliary glands, although there is still controversy as to whether the cellular origin of ICC is derived from progenitor cells, hepatocytes, or cholangiocytes [26,27,28]. We found unusual macroscopic and histopathological lesions of both types of ICC, small- and large-duct type, in the sandhill crane.

In humans, HCC is the most common primary liver malignancy. Hepatitis B and C viruses are independent risk factors for the development of HCC and ICC [29,30,31], and chronic liver disease usually progresses through cirrhosis to liver malignancy. Other pathways to developing HCC and ICC are through exposure to co-carcinogens such as toxins, as well as genetics and environmental factors [30]. From the metagenomics analysis conducted to identify the infectious etiology of the cholangiocarcinoma in the sandhill crane, no virus was found to be the cause of the tumors (e.g., avian hepatitis E virus or unveiled oncogenic viruses). However, it is presumed that the primary pathogen or chemical toxins were present since the bird showed severe cirrhosis of the liver.

Metagenomics revealed *Cutibacterium acnes* and *E. coli* in liver carcinoma of thesandhill crane with an abundant read count, although the bacteria were not detected in the histology and no culture was performed. *Cutibacterium* sp., a human skin-resident bacterium, is not known to be a pathogen found in veterinary science previously. However, it was reported as a pathogen found in a liver abscess in a person who underwent hepatectomy for HCC and was considered in delayed recovery after surgery [32,33]. It is well known that there is an increased abundance of *E. coli* in patients with liver cirrhosis and chronic inflammation, and *E. coli* may contribute to the process of hepatocarcinogenesis in humans [34]. Even if bacterial infection by *Cutibacterium* sp. identified from the ICC of the sandhill crane was an opportunistic or secondary infection together with *E. coli*, it may have induced an inflammatory process in its liver, and it seems likely that it played a role in exacerbating the cirrhosis associated with hepatic fibrogenesis. The inflammatory process may also be a reactive change secondary to tissue damage by neoplastic cells [35,36,37].

## 5. Conclusions

This is the first report where the anatomopathological characteristics of ICC in a sandhill crane are described. No viral cause was identified in the metagenomics analysis and PCR of the cholangiocarcinoma sample; however, a contributing role of *Cutibacterium* sp. and *E. coli* identified from metagenomics could not be excluded.

## Figures and Tables

**Figure 1 animals-13-03469-f001:**
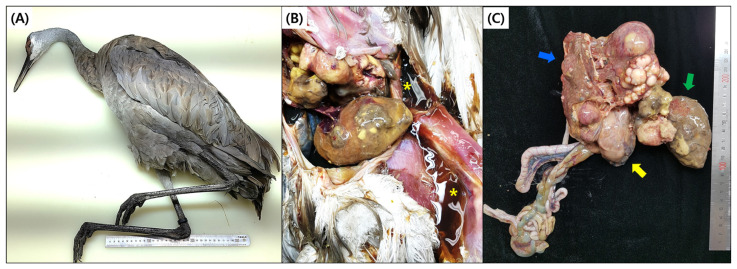
Gross lesions of liver carcinoma. (**A**) Carcass of sandhill crane, (**B**) ascites (asterisk) retained in the coelomic cavity and masses of pleomorphic neoplasm, and (**C**) cholangiocarcinoma, right liver (blue arrow). Gizzard (yellow arrow), left liver (green arrow).

**Figure 2 animals-13-03469-f002:**
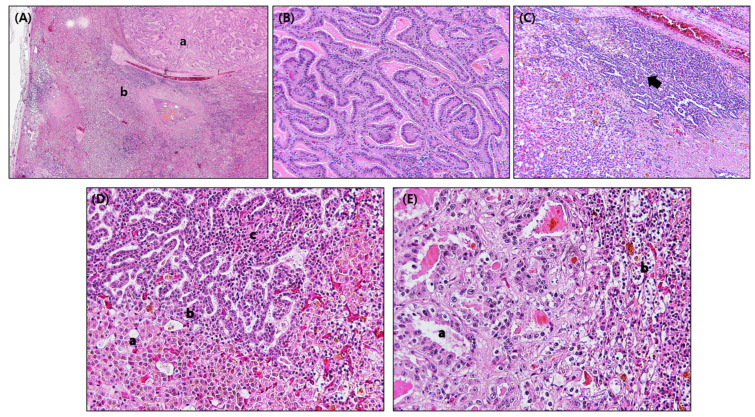
Histopathologic lesions of intrahepatic cholangiocarcinoma (ICC) in a sandhill crane. (**A**) Two compartments, one area with neoplastic cells similar to bile duct epithelium (a) and the other area with solid neoplastic columnar-cells containing only small luminal slits admixed with vacuolated hepatocytes and hemorrhages (b). (**B**) Large-duct-type ICC characterized by the mucin-producing papillary tumors developed through diffuse papillary proliferation of bile duct cells. (**C**) Moderately differentiated small-duct-type ICC is prominent, with irregular ductular patterns of neoplastic cells (arrows) that infiltrate the adjacent hepatic parenchyma. Fibrin deposition, severe hemorrhage, and hemosiderosis are shown. (**D**) Small-duct-type ICC that is composed of non-mucin-producing cuboidal cells, such as bile ducts (b) with hyperchromatic nuclei and condensed cytoplasm and deposition of fibrin between tumor cells (c), replacing normal hepatocytes (a). (**E**) A higher-magnification image showing large-duct-type ICC that is composed of columnar and mucin-producing cells characterized by large nuclei and abundant cytoplasm (a), compared to small-duct-type ICC (b). Original magnification: 40× for (**A**), 200× for (**B**,**C**), and 400× for (**D**,**E**).

**Table 1 animals-13-03469-t001:** Summary of metagenomic sequencing of intrahepatic cholangiocarcinoma sample with chicken liver as a control.

	Intrahepatic Cholangiocarcinoma Sample		SPF Chicken Liver (Control)	
	No. of Reads	% Trimmed Reads (% Bacteria)	No. of Reads	% Trimmed Reads (% Bacteria)
Raw Reads	13,851,588	-	14,527,223	-
Trimmed Reads	13,067,715	-	13,914,506	-
Host Reads	8,461,396	64.8	13,385,586	96.2
Eukaryota	238,614	1.8	5169	<0.1
Bacteria	2,194,087	16.8	10,648	0.1
*Cutibacterium acnes*	388,437	(17.7)	-	-
*Escherichia coli*	288,422	(13.1)	-	-
*Shigella flexneri*	195,335	(8.9)	-	-
*Chlamydia abortus*	140,030	(6.4)	-	-
*Hyphomonadaceae bacterium*	-	-	1844	(17.3)
*Firmicutes*	-	-	1700	(16.0)
Others	1,181,863	(53.9)	7184	(67.5)
Virus	4216	<0.1	668	<0.1
Phage (*Siphoviridae*)	4154		23	
Equine infectious anemia virus	58		-	
Fowl aviadenovirus E	4		-	
Chicken chapparvovirus	-		645	
Others	24,546	0.2	487	<0.1
Unclassified	2,144,856	16.4	511,948	3.7
Total	4,606,319	100	528,920	100

## Data Availability

The data generated or analyzed during this study are available from the corresponding author on reasonable request.

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
