# Peer review of "Intrahepatic Cholangiocarcinoma Identified in a Zoo-Housed Sandhill Crane (Grus canadensis): An Anatomopathological and Metagenomic Study"

_animals, 2023, doi:10.3390/ani13223469_

Round 1

Reviewer 1 Report

Comments and Suggestions for Authors

This article is a case report that describes a case of intrahepatic cholangiocarcinoma in a sandhill crane, focusing on histopathology and metagenomic analyses to identify potential etiologic agents.  It provides a description of a known tumor type in an uncommon species, describing the gross and histopathologic features of the tumor, and describes a metagenomic method used to identify infectious agents that may contribute to carcinogenesis in this species. 

Overall, the authors have adequately addressed my initial comments.  I have a few new comments that relate mainly to clarity. In lines 111-116, a reference to Fig 2A would be helpful in the description of the histopathology of the lesion.  In line 109 in the caption for Fig 2, “abdominal cavity“ should be replaced with “coelomic cavity”. Line 115-118 describing the perihepatitis, it may be preferrable to state that this inflammation is suspected to be due to bacterial infection since the presence of bacteria was not confirmed histologically or by culture.

Comments on the Quality of English Language

Overall, the English language quality is good.  There is some phrasing and organization that could be improved to help with clarity and flow.

Author Response

# Response to Reviewer's comments

In lines 111-116, a reference to Fig 2A would be helpful in the description of the histopathology of the lesion.

→ According to reviewer’s comments, we referred to the “Diagnostic histopathology of tumors, third ed. Christopher Fletcher” and modified the description of the histopathology of the lesion, line 126-131.

 In line 109 in the caption for Fig 2, “abdominal cavity“ should be replaced with “coelomic cavity”.

→ According to reviewer’s comments, the word was revised, line 124.

 Line 115-118 describing the perihepatitis, it may be preferrable to state that this inflammation is suspected to be due to bacterial infection since the presence of bacteria was not confirmed histologically or by culture.

→ According to your and other reviewer’s comments, the phrase was deleted, line 132.

Reviewer 2 Report

Comments and Suggestions for Authors

Thank you for this interesting case, unfortunately the cause of the tumor has not been conclusively clarified and I think the bacterial cause is more a assumption.

Line 41: The causes of tumors mentioned here refer more to poultry such as chickens, but there are other causes that should be mentioned for species that reach a high age.

Line 57: Which organs were used for virus isolation?

Line 101: Was the ascites examined, cell count, total protein, specific gravity, cytology?

Line 102: The question is whether the animal died of heart failure and the liver was just an incidental finding? Blood liver associated values before the death of the animal would have been interesting.

Line 117: But no microbiological culture was performed?

Line 132: Please add the organs which have been histologically examined.

Line 145: Cutibacterium acne is not specific pathogen. A microbiological culture would have been interesting to see if the bacterium is capable of reproducing and if other bacteria could have been grown.

Line 209: But were specific PCRs performed for this and other viruses, or just metagenomic analysis?

Lin 225: But the reason for the tumor were not conclusively found in the present study.  The question is whether further diagnostics such as microbiological cultivation, specific PCRs and the examination of other organ samples would have led to different results?

Author Response

# Response to Reviewer's comments

Line 41: The causes of tumors mentioned here refer more to poultry such as chickens, but there are other causes that should be mentioned for species that reach a high age.

→ According to reviewers’ comment, we revised the sentences, line 48-53.

Line 57: Which organs were used for virus isolation?

→ According to reviewer’s comment, we added a sentence, line 71.

Line 101: Was the ascites examined, cell count, total protein, specific gravity, cytology?

→ Before the necropsy, there were no finding suspicious for ascites, such as abdominal distension. So preparations and collection could not be made in advance to examine for ascites.

Line 102: The question is whether the animal died of heart failure and the liver was just an incidental finding? Blood liver associated values before the death of the animal would have been interesting.

→ According to reviewer’s comment, we added the words “and heart failure may be considered the cause of death in this bird.“ in line 115. In addition, there were no blood samples because the necropsy was submitted after the animal had already died.

Line 117: But no microbiological culture was performed?

→ Microbiological culture was not performed, because bacterial infection was not considered at the time of necropsy by mistake.

Line 132: Please add the organs which have been histologically examined.

→ According to reviewer’s comment, we added the organs in line 78 and line 148.

Line 145: Cutibacterium acne is not specific pathogen. A microbiological culture would have been interesting to see if the bacterium is capable of reproducing and if other bacteria could have been grown.

→ Unfortunately, microbiological culture was not performed

Line 209: But were specific PCRs performed for this and other viruses, or just metagenomic analysis?

→ According to reviewer’s comment, we added methods and results for viruses in line 85-91 and line 151-154. In fact, we had performed tests for virus isolation and detection but omitted for the sake of a brief case report. Because there was no positive result.

Lin 225: But the reason for the tumor were not conclusively found in the present study.  The question is whether further diagnostics such as microbiological cultivation, specific PCRs and the examination of other organ samples would have led to different results?

→ Thank you for your detailed comments. Although bacterial culture was not performed, we added the results of the virus PCRs. Conclusions about the cause of tumors presented in this study are unlikely to be affected.

Reviewer 3 Report

Comments and Suggestions for Authors

The manuscript entitled "Histopathology and Metagenomics of intrahepatic cholangioarcinoma identified in a sandhill crane (Grus canadensis) in a zoo" was evaluated. Gross and histological characteristics of ICC in a sandhill crane are first reported, and the cause of death was investigated.This manuscript helps to understand histological characteristics of ICC in avain. The paper is in the scope of the journal.

Negative aspects

line 66-69: residual bacterial and debris from tissue sample was removed, whether this process would eliminate the potential etiology pathogen of this disease?

Line 72: “1㎖”should be replaced with “1ml”.

line 212-221:  whether E.coli is associated with ICC should be discussed as Cutibacterium acnes.

Author Response

# Response to Reviewer 3's comments

line 66-69: residual bacterial and debris from tissue sample was removed, whether this process would eliminate the potential etiology pathogen of this disease?

The pointed-out phrase out was deleted for a brief description of method according to another reviewer’s comment. In this pretreatment, most of what was removed were host cells. Since the bacterial nucleic acids are amplified during the subsequent experiment, the bacterial pathogen of this disease can be identified. A few studies(https://doi.org/10.1186/s12917-023-03732-y, https://doi.org/10.3390/vetsci9070332) using this method show that metagenomics results and bacteria cultured using traditional method are consistent.

Line 72: “1㎖”should be replaced with “1ml”.

The pointed-out unit was deleted for a brief description of method according to another reviewer’s comment.

line 212-221:  whether E.coli is associated with ICC should be discussed as Cutibacterium acnes.

According to reviewer’s comment, we added the sentences discussing that E. coli is associated with hepatic neoplasm in line 243-245.

Reviewer 4 Report

Comments and Suggestions for Authors

Dear authors

Major editing is recommended to improve the reviewed manuscript, as follows

L 2-4 The title could be reorganized, an option: Intrahepatic cholangiosarcoma in a zoo-housed sandhill crane (Grus canadensis): Anatomopathological and metagenomic features

L9 Please include the scientific name of the studied bird

L11 Despite the age of the studied bird is unknown, the bird can be characterized as “adult”.

L16 Is E. coli excluded from the potential role triggering the hepatic tumor?

L17-19 and L40-41 Please rewrite this sentence because only a limited range of tumors can be caused by viral infections such as Marek’s disease (T-cell lymphoma) and avian leukosis-sarcoma group (B-cell lymphoma and others). A big group of tumors spontaneously affecting a wide range of avian species are not related with infectious agents.

L18 and L41 Please consider that “leukosis” is the correct term instead of “leucosis”.

L21 Please improve and rewrite this sentence to describe the gross aspect of the hepatic tumor.

L27-29 Are Cutibacterium spp. and E. coli both considered as etiologic agents of this hepatic tumor? Did you discard the spontaneous presentation of this tumor?

L33 Please include the geographical origin of this bird and the current conservation status.

L38-40 Please expand the etiologic infectious and toxic causes of death affecting sandhill cranes and include age and life conditions of these birds

L45-47 Please rewrite the aim of the study in order to provide a concise sentence.

L51 Please rewrite this paragraph. An option could be: “A carcass of a male, adult, captive sandhill crane, was submitted from a zoo located at …. for postmortem examination and diagnostic work-up to … in November 2021. The bird, which was imported from…, belonged to a crane restoration project in…”

L59 Please include the names of the organs within “spectrum” of tissues sampled during the postmortem examination. Did you perform bacteriologic culture on the affected liver?

L65 to L87 The length of this paragraph describing methodology need to be addressed, please replace it by a brief paragraph.

L101 Regarding the gross evaluation of this bird                                               - Did you notice a distended abdomen in this bird? How severe was the ascites in this bird? Did you measure the amount of ascitic fluid?                    - Did you measure the hepatic neoplastic mass? Were the nodules from the same size? Were they firm, soft o cystic?                                                            - Can you expand the term cirrhosis to characterize the affected liver? Did you see the hemorrhages on the liver or within the abdominal cavity?

L108 to L110 Please include the scientific name of the bird, the complete name of the neoplasia, and clarify the references of the neighboring organs by arrows or asterisks to clarify the anatomic references in the Figs 1B and 1C.

L111 to L125 Regarding the histopathologic findings of this neoplasia            - The terms “low magnification”, “high magnification” and “higher magnification” can be deleted.  “Histopathologic” can be used instead of “histopathological”.                                                                                            - Please include the name of the neoplastic entity at the end the paragraph only after finishing with the histopathologic description.                                  - L116-117 Please delete “which appear to be caused by bacterial infection”

L182 Did you mention the other organs different from liver which were sampled for histopathologic evaluation?

L143-144 The name of the neoplasia can be replaced by “ICC” at this point

L154 to L168   Caption of microscopic figures                                                  - The terms “low magnification”, “high magnification” and “higher magnification” can be deleted.  “Histopathologic” can be used instead of “histopathological”.                                                                                             -  L162 Is it correct to include “in the interstitial spaces”?

L 171-172 A reference needs to be added to support the malignancy of tumors found in Gruidae species.     

L175-177 Please rewrite this sentence to include both gross and histopathologic as “anatomopathological”.

L177-180 Please clarify which of the named bird species were affected by cholangiocarcinoma and if they were free-ranging, zoo-housed or domestic.   Scientific names of these birds need to be included also.

L183-185 Please rewrite the sentence to clarify it.

L199-201 “Uncertain roles of toxins and unknown environmental factors” are not clarifying the potential trigger factors or potential etiologies in this case. Please delete this sentence.

L219 to 221 Please include a reference to support this sentence.

L225-226 Did you exclude E. coli role? If yes, please explain why.

Round 2

Reviewer 2 Report

Comments and Suggestions for Authors

Thank you for revising the manuscript. In the course of the text and in the conclusion, it should be added that no viral cause could be found in the metagenomics and in the PCR. Line 28 “its potential infectious ethology. Line 142: missing space. Lines 220 to 230: it should be added that the bacteria were not detected in the histology and no culture was performed. Otherwise, I have no further comments.

Author Response

In the course of the text and in the conclusion, it should be added that no viral cause could be found in the metagenomics and in the PCR.

→ According to reviewer’s comments, we added the words, line 15, 27 and 242.

Line 28 “its potential infectious ethology.

→ According to reviewer’s comments, we added the word, line 30.

Line 142: missing space.

→ According to reviewer’s comments, we added “space”, line 148.

Lines 220 to 230: it should be added that the bacteria were not detected in the histology and no culture was performed. Otherwise, I have no further comments.

→ According to reviewer’s comments, we added the phrase, line 226-227.

Reviewer 4 Report

Comments and Suggestions for Authors

Dear authors,

Minor editing is suggested, as follows

L46 Please use “genus” instead of “family”.

L55 Please clarify if bacteriologic studies from liver or other organs were performed. This could be a good addition to link with the metagenomic result detecting Cutibacterium sp. and E. coli in the affected neoplastic liver.

L66-67 Please clarify that the allantoid fluid obtained from SPF embryonating eggs inoculated with aliquots of homogenized trachea, cecal tonsils, kidney and liver samples were tested against avian influenza virus and Newcastle disease virus by hemagglutination tests.

L79 Please reorder the beginning of the sentence as “… A liver sample from a healthy chicken…”

L108 Please use “ascitic fluid” instead of “ascites”

L112 The location of “hemorrhages” described in the liver need to be included. Were these hemorrhages were noted? On the hepatic capsule or on the cutting surface? How were distributed? Diffusely? Multifocal? Focal extensive? How severe were?

L114-115 Can you please explain that the hepatic neoplasia of this case was mainly replacing the liver and no remaining liver area was noted.

L124 Please clarify why you included “pseudocapsule”.

L125 Please use the term “…. Intralesional bacterial colonies…” instead of “… Bacteria…”

L141 Please consider that metastasis cannot be discarded in this case. You did not include key tissues to the histopathologic evaluation of metastasis such as lungs, bone marrow and brain. These organs are usually affected by metastasis of malignant neoplasia in avian and mammal species.

L142-143 Please rewrite this paragraph to clarify it.

L183, L193 and L222 The term “previously” could be added at the end of the three sentences.

L184 Please use the term “case report” instead of “case study”.

L197 Please use “avian tumors” instead of “avian pathology”.

L209-L210 Please do not discard the presence of metastasis in this case because no brain, lung or bone marrow samples were included for the histopathologic study of the sandhill crane.

L209 and L233

L224 to L226 Please clarify if this paragraph is refereed to an animal or human case.

L233 to L236 Please improve this paragraph.

Author Response

L46 Please use “genus” instead of “family”.

→ According to reviewer’s comments, we revised the word, line 48.

L55 Please clarify if bacteriologic studies from liver or other organs were performed. This could be a good addition to link with the metagenomic result detecting Cutibacterium sp. and E. coli in the affected neoplastic liver.

→ According to reviewer’s comments, we added the sentence, line 74-75.

L66-67 Please clarify that the allantoid fluid obtained from SPF embryonating eggs inoculated with aliquots of homogenized trachea, cecal tonsils, kidney and liver samples were tested against avian influenza virus and Newcastle disease virus by hemagglutination tests.

→ According to reviewer’s comments, we revised the sentence, line 68-70.

L79 Please reorder the beginning of the sentence as “… A liver sample from a healthy chicken…”

→ According to reviewer’s comments, we added the phrase, line 82-83.

L108 Please use “ascitic fluid” instead of “ascites”

→ According to reviewer’s comments, we revised the words, line 111.

L112 The location of “hemorrhages” described in the liver need to be included. Were these hemorrhages were noted? On the hepatic capsule or on the cutting surface? How were distributed? Diffusely? Multifocal? Focal extensive? How severe were?

→ According to reviewer’s comments, we added the sentence, line 117-118.

L114-115 Can you please explain that the hepatic neoplasia of this case was mainly replacing the liver and no remaining liver area was noted.

→ According to reviewer’s comments, we added the phrase, line 119-120.

L124 Please clarify why you included “pseudocapsule”.

→ According to reviewer’s comments, we added the phrase, line 129-130.

L125 Please use the term “…. Intralesional bacterial colonies…” instead of “… Bacteria…”

→ According to reviewer’s comments, we revised the words, line 131.

L141 Please consider that metastasis cannot be discarded in this case. You did not include key tissues to the histopathologic evaluation of metastasis such as lungs, bone marrow and brain. These organs are usually affected by metastasis of malignant neoplasia in avian and mammal species.

→ According to reviewer’s comments, we deleted the sentence related with metastasis, line 146.

L142-143 Please rewrite this paragraph to clarify it.

→ According to reviewer’s comments, we revised the sentence, line 149.

L183, L193 and L222 The term “previously” could be added at the end of the three sentences.

→ According to reviewer’s comments, we added the word, line 190, 199 and 228.

L184 Please use the term “case report” instead of “case study”.

→ According to reviewer’s comments, we revised the word, line 190.

L197 Please use “avian tumors” instead of “avian pathology”.

→ According to reviewer’s comments, we revised the word, line 203.

L209-L210 Please do not discard the presence of metastasis in this case because no brain, lung or bone marrow samples were included for the histopathologic study of the sandhill crane.

→ According to reviewer’s comments, we deleted the sentence related with metastasis, line 215.

L209 and L233

L224 to L226 Please clarify if this paragraph is refereed to an animal or human case.

→ According to reviewer’s comments, we added the word, line 223.

L233 to L236 Please improve this paragraph.

 → According to reviewer’s comments, we revised the paragraph, line 240-243.